# Risk Assessment of Indoor Air Quality and Its Association with Subjective Symptoms among Office Workers in Korea

**DOI:** 10.3390/ijerph19042446

**Published:** 2022-02-20

**Authors:** Dayoung Jung, Youngtae Choe, Jihun Shin, Eunche Kim, Gihong Min, Dongjun Kim, Mansu Cho, Chaekwan Lee, Kilyong Choi, Byung Lyul Woo, Wonho Yang

**Affiliations:** 1Department of Occupational Health, Daegu Catholic University, Gyeongsan 42472, Korea; ekdud37@korea.kr (D.J.); kickilbo@naver.com (Y.C.); shinjs1130@naver.com (J.S.); kimeunchae77@gmail.com (E.K.); alsrlghd000@naver.com (G.M.); jun961123@gmail.com (D.K.); s100002@naver.com (M.C.); 2Environmental Health Research Division, National Institute of Environmental Research, Incheon 22733, Korea; 3Institute of Environmental and Occupational Medicine, Medical School, Inje University, Busan 47392, Korea; lck3303@hanmail.net; 4Department of Environmental Energy Engineering, Anyang University, Anyang 14028, Korea; bestchoi94@anyang.ac.kr; 5Industrial Hygiene, Preventive Medicine, Force Health Protection, U. S. Army Medical Department Activity-Korea/65th Medical Brigade, Unit # 15281, APO (Army Post Office) AP (Armed Force Pacific) 96271-5281, USA; yissoyi@gmail.com

**Keywords:** indoor air quality, office workers, risk assessment, health effects, subjective symptoms

## Abstract

The 2014 Time-Use Survey of Statistics Korea revealed that office workers are increasingly spending more than eight hours at work. This study conducted an exposure assessment for office workers in Korea. Indoor and outdoor air pollutants were measured in offices. A self-administered questionnaire was employed to determine work information, indoor air quality (IAQ) awareness, and subjective symptoms for 328 workers. Indoor air concentrations for measured air pollutants were below IAQ guideline values. The average concentrations of target air pollutants did not show significant differences except for benzene, which had relatively a higher concentration in national industrial complexes. The indoor benzene, ethylbenzene, and acetaldehyde concentrations were higher in offices where workers were having dry eye, ophthalmitis, and headache symptoms. This study provides reference values to manage IAQ in offices, suggesting that if the benzene concentration exceeds 4.23 μg/m^3^ in offices, it could cause dry eye symptoms. Considering the increasing working hours for office workers and health effects, workers’ exposure to indoor pollutants should be reduced. In addition, the IAQ was heavily influenced by outdoor air levels and various indoor sources. Therefore, in areas with relatively high air pollution, greater monitoring and management is required considering the influence of outdoor air quality.

## 1. Introduction

Indoor air quality (IAQ) is the air quality inside buildings. As most people spend 88% of their time indoors, the management of indoor environments has emerged as an important factor that determines the quality of life [1]. The IAQ is an indicator of exposure to indoor air pollutants.

According to the results of a Time-Use Survey in 2014 by Statistics Korea, office workers spend more than eight hours a day in the office [2]. The number of office workers was the largest among the nine standard occupational categories and has shown a steadily increasing trend over the past few years [3]. In addition, office workers may comprise sensitive receptors, namely pregnant women, the elderly, and more vulnerable populations such as individuals with chemical hypersensitivity and those who are immunocompromised [4]. Furthermore, because the indoor environment is complex, the cumulative effect of thermal conditions, IAQ, electromagnetic fields, and environmental conditions should be considered [5].

IAQ in office environments can result in diverse disease incidence for workers and may be associated with subjective symptoms such as headaches, fatigue, sick building syndrome (SBS), along with non-specific symptoms affecting the eyes and nose [6,7,8,9,10,11]. SBS symptoms include general symptoms (headache, fatigue, difficulty concentrating), mucosal symptoms (eye, throat, nose irritation, cough), skin symptoms (dry or flushed facial skin, itchy eyes, or itchy ears), and asthma and asthma-like symptoms [12,13]. Health effects may show up years after exposure or after long or repeated periods of exposure. These effects, which include respiratory diseases, heart disease, and cancer, can be severely debilitating or fatal.

Most countries manage IAQ based on policy guidelines, and only a few countries have legislation for office IAQ, including Japan, Australia, Singapore, and Malaysia [14]. The Ministry of Employment and Labor in Korea is implementing guidelines to ensure the appropriate maintenance of IAQ in accordance with the Occupational Safety and Health Act. Similarly, the American Society of Heating, Refrigerating, and Air Conditioning Engineers (ASHRAE) established environmental standards regulating ventilation and IAQ to improve air quality during building design and construction [15]. While studies on the effects of common indoor air pollutants on the health of office workers indicate many harmful effects, there is considerable uncertainty about the concentrations or periods of exposure that produce specific health problems [16]. Further research is needed regarding indoor pollutant characteristics and the influence of the outdoor environment.

This study aims to evaluate indoor air pollutant concentrations, thermal conditions, subjective symptoms, cancer risk, and exposure characteristics according to various internal and external office environments, and to provide reference data for creating a healthy and comfortable indoor environment.

## 2. Materials and Methods

### 2.1. Target Office Buildings

This study was conducted from July to September, 2016. Target offices were classified based on three categories. After selecting suitable regions for the three categories, 10 offices in each group were randomly recruited. In Group 3, 11 offices were selected, and the study was conducted for a total of 31 offices. Group 1 was offices in metropolises such as Seoul, Incheon, Daegu, and Busan and included hospitals, schools, and banks. Group 2 was company offices on sites designated as national industrial complexes such as Ulsan, and Group 3 was offices in buildings separated from the factory within the company premises. Seoul is the capital of South Korea. Incheon, Daegu, and Busan are the biggest metropolitan cities in Korea. Ulsan is a city designated as a national industrial complex or industrial district and includes a petrochemical complex.

### 2.2. Indoor and Outdoor Air Quality Measurements

The air pollutants for 31 office indoor and outdoor environments were assessed. The temperature, humidity, and target air pollutants are shown in Table 1. Target air pollutants were carbon monoxide (CO), carbon dioxide (CO_2_), formaldehyde (HCHO), acetaldehyde, total volatile organic compounds (TVOCs), nitrogen dioxide (NO_2_), and ozone (O_3_). CO, CO_2_, HCHO, acetaldehyde, and TVOCs were measured according to the air quality guidelines for offices as per the Ministry of Employment and Labor, Korea, and using the IAQ testing method. The measurement of O_3_ was in accordance with the Korea Occupational Safety and Health Agency (KOSHA) CODE A-1-2004 method, and NO_2_ was measured using a badge-type diffusive sampler [17].

The sampling was conducted during working hours (9 am–6 pm). Calibration of the measuring instrument was entrusted before and after measurement. The measuring instruments were placed in safe locations on a flat surface at a height of 1–1.5 m to simulate the breathing zone and located roughly 1 m away from walls, doors, or air conditioning units to minimize the impact on the airflow path and source. The outdoors was in front of a door with a shelter cover to protect from rain or wind. 

### 2.3. Questionnaire and Checklist

A total of 328 self-administered questionnaires were distributed to workers in 31 offices on the day of measurement, which covered individual working conditions (working year, daily working hours), IAQ awareness (work efficiency, occurrence of health effects, stress), and subjective symptoms (headache, nasal congestion, eye inflammation). Basic information on the office building (address, construction year) and work environment (ventilation, air purifier, air conditioning system) was recorded using a checklist.

### 2.4. Health Risk Assessment

A health risk assessment was conducted to quantify the cancer risk and hazard quotient of carcinogens and non-carcinogens for the workers, respectively. The exposure time from the office workers’ questionnaire survey was used. An exposure frequency of 250 days per year and exposure duration of 35 years were assumed, as shown in Appendix A. The cancer slope factor (SF), reference concentrations, and reference dose factors (RfD) were derived from the United States (US) Environmental Protection Agency (EPA) Integrated Risk Information System (IRIS) database [18]. We applied a reference dose concentration of NO_2_ and O_3_ of 0.06 ppm (0.03 mg/m^3^), a 24 h average standard for atmospheric environment and respiration rate (14.3 m^3^/day), a body weight of 64.2 kg, and the average life expectancy of adults, based on the Korean Exposure Factors Handbook [19].

The cancer risk (CR) of carcinogens was calculated by multiplying lifetime average daily doses (LADDs) with the SF. The HQ of non-carcinogens was calculated by dividing average daily doses (ADDs) by reference dose (US EPA, 1989, 1987).
(1)$$\mathrm{LADDs}\;\left(\mathrm{mg}/\mathrm{kg}/\mathrm{day}\right)=\frac{C\;\times \;\mathrm{IR}\;\times \;\mathrm{ED}\;\times \;\mathrm{EF}\;\times \;\mathrm{ET}}{\mathrm{BW}\;\times \;\mathrm{AT}\;\times 1000\times 24}$$
(2)$$\mathrm{ADDs}\;\left(\mathrm{mg}/\mathrm{kg}/\mathrm{day}\right)=\frac{C\;\times \;\mathrm{IR}}{\mathrm{BW}}\;$$
(3)$$\mathrm{ECR}=\mathrm{LADDs}\left(\mathrm{mg}/\mathrm{kg}/\mathrm{day}\right)\times \;\mathrm{Slope}\;\mathrm{factor}\left({\left(\mathrm{mg}/\mathrm{kg}/\mathrm{day}\right)}^{-1}\right)$$
(4)$$\mathrm{HQ}=\frac{\mathrm{ADDs}\left(\mathrm{mg}/\mathrm{kg}/\mathrm{day}\right)}{\mathrm{RfD}\left(\mathrm{mg}/\mathrm{kg}/\mathrm{day}\right)}$$
where LADDs is the lifetime average daily potential dose rate in mg/kg-day; ADDs is the average daily potential dose rate (mg/kg/day); EF is the exposure frequency for the exposed individual (day/year); ED is the exposure duration for the exposed population, in years; and AT is the amount of time over which exposure is averaged, in days (70 years = 25,550 day/year). ET is the total daily time during which the population is exposed to air pollutants in the course of a work period, usually hours per day. C is the air concentration in offices (μg/m^3^); IR is the inhalation rate typically expressed in liters of air inhaled per hour (m^3^/day); and BW is the body weight of an individual, typically expressed in kilograms (kg). Excess cancer risk (ECR) is the probability that exposure to a hazardous air pollutant has a carcinogenic effect in one or more individuals.

A deterministic risk assessment was undertaken using central tendency exposure (CTE; mean or the 50th percentile); the probabilistic risk assessment is presented with mean, minimum, maximum values, 25%, 50%, 75%, 90%, and 95% by Monte-Carlo simulations using the @RISK (Palisade Software, Ltd.) program.

### 2.5. Statistical Analysis

Statistical analysis was performed using SPSS ver. 21 (IBM Company, New York, NY, USA), as the limit criterion of statistical significance assumed a *p*-value < 0.05 (95% reliability). The measured concentration values were determined for normality using Kolmogorov–Smirnov analysis. The statistical analyses of air concentration, questionnaires, and checklists were conducted using descriptive statistics, cross-tabulation analysis, independent *t*-tests, one-way analysis of variance (ANOVA), and stepwise multi-linear regression analysis.

A stepwise multi-linear regression analysis was conducted to determine statistically significant variables affecting indoor air pollutant concentrations. Variables included humidity, temperature, outdoor measurement concentration, number of computers, printers, and photocopiers, number of workers, age of the building, remodeling, and use of air purifiers. Dummy variables were used for the age of the building (more than 10 years (1), less than 10 years (0)), remodeling (with (1), not (0)), and whether air purifiers are used (used (1), not used (2)).

## 3. Results

### 3.1. Questionnaire and Checklist for Office Workers

#### 3.1.1. Office Characteristics

The average number of workers in the 31 offices was 28, as shown in Appendix A. The average number of working hours was 8.19 ± 2.90. The number of computers, printers, and copiers was 24.2 ± 27.1, 3.2 ± 4.9, and 4.0 ± 5.7, respectively. The numbers of offices with indoor charcoal or plant pots, ventilators, and air purifiers were 24 (7.34%), 14 (45.2%), and 9 (29%), respectively. The average frequency of ventilation per day with opening and closing of windows was 1.92 ± 2.56 times, and the average duration of ventilation was 27.17 ± 4.51 min. A total of 15 out of 31 buildings (48.4%) had air conditioning equipment installed. The characteristics of each office are shown in Appendix A. The number of workers in Group 3 was 31.00 ± 41.25, more than that of Group 1 (28.80 ± 19.83) and Group 2 (23.70 ± 21.01). The office floor areas were 190.01 ± 137.76 m^2^, 236.54 ± 126.55 m^2^, and 339.91 ± 445.17 m^2^ in Group 1, 2, and 3, respectively. The office volume of Group 3 was the largest at 894.77 ± 1160.16 m^3^. Group 1 was closer to the street than other groups (Appendix A).

#### 3.1.2. Association between Subjective Symptoms and Environmental Factors

Figure 1 depicts the results of the office workers’ perception survey. For the “main reason for indoor air pollution” question, 47.4%, 35%, 9.8%, 5.5%, and 2.3% of the respondents mentioned insufficient ventilation, indoor sources, external air inflow, workplaces (factory), and others, respectively. With regard to “the degree of health-related effects impacted by IAQ” question, 1.6%, 17.9%, 59.2%, 12.9%, and 8.4% of respondents believed that it was very serious, serious, moderately, slightly, and not at all serious, respectively. For “the degree of decline in work efficiency caused by IAQ” question, 0.9%, 42%, 56.2%, 0.3%, and 0.6% of the respondents felt that it was very bad, bad, never occurred, good, and very good, respectively. With regard to the “degree of stress from the amount of work” question, 1.9%, 16.9%, 51.9%, 21.6%, and 7.7% of the respondents felt that it was very serious, serious, moderately, slightly, and not at all serious, respectively.

Figure 2 shows the self-diagnosis experience and pseudo-diagnosis rates for participants’ subjective symptoms. During working hours, workers frequently reported fatigue (22.12%) and ophthalmitis (21.20%) and occasionally reported headache (23.72%), sneeze (24.04%), ophthalmitis (25.00%), fatigue (25.64%), dryness of the throat (27.65%), and drowsiness (31.65%). The doctors’ diagnoses of these subjective symptoms were dry eye (18.56%), ophthalmitis (16.17%), and nasal stuffiness and rhinitis (15.85%).

Office workers who frequently reported dry eye, ophthalmitis, nasal stuffiness and rhinitis, and headache as subjective symptoms were classified into symptom group and non-symptom group. These groups were further analyzed based on the characteristics of the working environment (Table 2). In the group reporting subjective symptoms, males reported more dry eye, ophthalmitis, as well as longer working hours than females (*p* < 0.05). The group reporting headache symptoms was associated with younger age and more working hours (*p* < 0.05). Moreover, although not statistically significant, there were more females (38.70%) in the symptom group than in the non-symptom group (28.00%).

Table 3 shows the comparison of average air pollutant concentrations between the symptom group and non-symptom group, and only significant results are shown. Benzene, ethylbenzene, and acetaldehyde concentrations were significantly higher in the ophthalmitis symptom group than in the non-symptom group. Dry eye symptoms are associated with benzene concentration and were higher in the symptom group.

### 3.2. Indoor and Outdoor Air Pollutant Concentrations 

Regarding the concentrations measured in offices, the average indoor temperature and humidity were 26.74 ± 2.38 °C and 56.94 ± 11.51%, respectively, while the average outdoor values for the same were 28.58 ± 3.55 °C and 66.26 ± 11.06%, respectively, as shown in Appendix A. The 95% values of NO_2_ (0.06 ppm), TVOCs (1066.58 μg/m^3^), and HCHO (176.00 μg/m^3^) exceeded the standard values recommended by the Ministry of Labor, while the 95% values of toluene (265.30 μg/m^3^) exceeded the IAQ guideline values (260 μg/m^3^) recommended by the World Health Organization (WHO).

The indoor average concentration value of benzene (*p* = 0.048) showed a significant difference among the three office groups, with 4.48 ± 4.31 μg/m^3^ in Group 1, 5.23 ± 3.28 μg/m^3^ in Group 2, and 2.82 ± 4.03 μg/m^3^ in Group 3 as shown in Appendix A.

The TVOCs concentration of 373.54 μg/m^3^ in Group 3 was higher than that of Group 1 and 2 (364.20 μg/m^3^, 334.93 μg/m^3^); this revealed that the indoor-to-outdoor (I/O) concentration ratio of VOCs (toluene, ethyl benzene, *m, p*-xylene, *o*-xylene) and aldehyde (HCHO, acetaldehyde) was higher than that of other air pollutants. Notably, the I/O concentration ratio of HCHO and acetaldehyde in Group 2 was 15.40.

Indoor humidity showed a positive correlation with the NO_2_ level (*p* = 0.048) and a negative correlation with temperature (*p* = 0.052). The indoor O_3_ level showed a negative correlation with the NO_2_ level (*p* = 0.019). Outdoor temperature showed a positive correlation with TVOCs (*p* = 0.005) and the HCHO level (*p* = 0.019). The outdoor O_3_ level showed a negative correlation with the NO_2_ level (*p* = 0.009). The outdoor CO_2_ level showed a positive correlation with the O_3_ level (*p* = 0.023) and a negative correlation with the HCHO level (*p* = 0.004). The levels of benzene, toluene, ethylbenzene, *m, p*-xylene, and *o*-xylene were correlated with both indoors and outdoors (*p* < 0.01) in Appendix A.

The factors affecting indoor air pollutant concentrations are shown in Table 4. The results of a linear regression model between indoor air pollutants’ concentrations and outdoor air pollutants’ concentration, temperature, and humidity revealed that outdoor pollutants excluding CO, toluene, temperature, and humidity had a significant effect on indoor pollutants’ concentration (*p* < 0.05).

The benzene level was affected by outdoor humidity, HCHO, and benzene concentration, as well as photo copiers; the coefficient of determination (R^2^) for benzene was 0.626, showing a strong relationship among air pollutants. The toluene, ethylbenzene, and *m, p*-xylene levels were affected by photo copiers, and the TVOCs level was affected by printers. The *o*-xylene level was affected by the HCHO outdoor concentration. The HCHO and acetaldehyde levels were affected by the use of air purifiers, and indoor HCHO and acetaldehyde concentrations were significantly lower in offices with air purifiers than those without by 1.784 and 1.048 μg/m^3^, as shown in Table 4.

### 3.3. Health Risk Assessment 

Table 5 depicts health risk assessment according to exposure to air pollutants in the office. The excess carcinogenic risk of benzene, HCHO, and acetaldehyde was 2.9 × 10^−6^, 4.8 × 10^−5^, and 2.7 × 10^−6^, respectively, which exceeded the carcinogenic risk criteria (1.0 × 10^−6^). Probabilistic evaluation values exceeded the risk criteria. Non-carcinogens were at a safe level that did not exceed the non-carcinogens’ risk standard 1 (HQ).

## 4. Discussion

Indoor air pollutants generated in factory workplaces are generally higher than in other office environments. In addition, several studies show that health risk, individual sensitivity, and subjective symptoms related to indoor air pollutants in the office are related to the duration of stay and show that the degree of discomfort decreases over time. The number of office workers in Korea has been continuously increasing for several years. Given that office workers spend much of their time outside of the home at the office, indoor air pollution should be managed to reduce health risks. As a result, this study attempts to source reference data based on office classification, including the characteristics and occurrence of subjective symptoms in relation to indoor air pollutant concentrations.

In this study, only one office had an air conditioning system that required filter cleaning. However, the satisfaction of office workers regarding IAQ did not differ from that of office workers in places where operation and management were not conducted regularly. For these offices, air quality was managed through ventilation. According to Zhang’s study (2017), when duct cleaning is not performed regularly, the HVAC system is not configured with the proper components; dust and oil components accumulate in the ducts, leading to operational problems. In addition, the air circulation rate decreased, and residents’ satisfaction was low [20]. Therefore, to manage indoor air conditioning better through the HVAC system, it is necessary to install a HVAC system that has high removal efficiency of indoor air pollutants mainly generated in the office to be managed and considers the characteristics of the building [21].

In the evaluation of indoor air pollutant concentrations, the concentrations of benzene, ethylbenzene, and aldehyde (HCHO, acetaldehyde) in Group 2 and 3 were high. In particular, the benzene concentration showed a significant difference among the three categories of groups. The study undertaken in Spain reported that benzene concentrations in industrial complexes were significantly higher [22]. In a similar study, air pollution index levels and industrial emissions were correlated with R^2^ values of 0.4791 [23,24]. Therefore, the effect of outdoor air quality on IAQ should be considered more seriously in industrial complexes such as petrochemical complexes [25].

Many studies reveal that IAQ in offices is affected by the emission of pollutants from indoor sources, including copiers, printers, toners, carpets, furniture, air fresheners, and cleaning supplies [26,27]. This study indicated that the O_3_ I/O concentration ratio was 0.75, and the outdoor and indoor R^2^ value for NO_2_ was 0.53. The indoor O_3_ and NO_2_ levels had a higher correlation with the outdoor air quality than with the indoor air quality. However, assuming that there is no indoor source, when modeling with 80% accuracy using the I/O ventilation rate and surface removal rate data, I/O for O_3_ was 0.47 with natural ventilation [28]. The indoor pollutant concentrations were also affected [29,30].

The levels of TVOCs, HCHO, and acetaldehyde had a strong association with outdoor air quality and indoor sources, such as air fresheners, building materials, and indoor furniture that emit HCHO and VOCs; the HCHO and acetaldehyde concentrations tended to decrease in older buildings [31]. The xylene and *n*-hexane concentrations significantly increased with the frequency of cleaning, and the benzene and toluene concentrations in offices using air fresheners and fragrances were significantly higher [32].

In summary, eye-related subjective symptoms (eye irritation and strain, dry eye, and itchiness) were the most frequently reported symptoms among office workers. Nasal congestion, rhinitis, and headache symptoms were also relatively high.

There was a significant association between HCHO and TVOCs and subjective symptoms such as eye irritation, swollen eyes, runny nose, and dry eye [16,33]. The benzene, ethylbenzene, and acetaldehyde levels were high in offices in the symptom group, and there was a significant association with subjective symptoms. The results of this study may be provided as a reference value for managing IAQ in the office, suggesting that should the benzene concentration exceed 4.23 μg/m^3^ in the office, dry eye symptoms may result.

The subjective symptoms associated with SBS differ based on demographic characteristics such as smoking status, age, gender, and diagnosis of asthma [34]. In this study, the group that reported subjective symptoms was younger and worked longer hours per day than the group that did not, and males were more sensitive than females (*p* < 0.05). Most of the symptom groups were two to three times more common in women than men, though recent studies showed that men were more sensitive than women in the office [35,36,37]. In a study that evaluated risk factors affecting SBS symptoms for office workers, the length of the workday is the most influential risk factor [38]. In an indoors home group reporting subjective symptoms due to aldehydes, the proportion of women is higher than that of men, presumably as women spend more time in the home than men, and the proportion of older respondents is higher (*p* < 0.01, *p* < 0.05) [39]. Therefore, the time spent indoors is a general characteristic that has the greatest influence on subjective symptoms. Upon evaluating the carcinogenic risk of benzene, HCHO, and acetaldehyde for office workers, HCHO was 4.8 × 10^−5^, indicating a higher carcinogenic risk than the other pollutants. These results were 2.86 times higher than the carcinogenic risk of 1.7 × 10^−5^ in new apartments [40]. Therefore, while the adverse impacts on health due to IAQ in offices may not generally result in incurable or acute diseases, there may be exposure to very harmful carcinogens.

A limitation of this study is that we could not evaluate the seasonal variation in the effect of outdoor air pollution on IAQ. In addition, the study was limited as individual measurement was not permitted owing to the noise generated by the measurement instruments and the impact on workers during working hours. Variation in exposure in humans showed a relatively high longitudinal correlation between the inter-individual mean and ambient level [41]. Accordingly, in general, air sampling in individual exposure assessment is used more widely on specific groups suspected of exposure. Another limitation was the time lag between the time of measurement and the onset of symptoms. However, the IAQ might be at similar levels for a certain period of time, and symptoms could be momentary rather than long-term.

## 5. Conclusions

The indoor air pollutants’ concentrations measured were below the IAQ guideline values of the Korean Ministry of Employment and Labor. However, indoor benzene, ethylbenzene, and acetaldehyde concentrations in offices had a significant association with dry eye, ophthalmitis, and headache symptoms, showing that concentrations were higher in offices with workers with subjective symptoms. Furthermore, the excess carcinogenic risk of benzene, HCHO, and acetaldehyde was 2.9 × 10^−6^, 2544.8 × 10^−5^, and 2.7 × 10^−6^, respectively, which exceeded the carcinogenic risk criteria (1.0 × 10^−6^). The probabilistic evaluation values of carcinogenic pollutants exceeded the risk criteria. This study suggests that exposure of office workers to air pollutants during office working hours should be reduced, particularly in light of the increase in numbers of office workers and increased overtime. The IAQ in offices was heavily influenced by outdoor air quality and several indoor sources. Therefore, in areas with a relatively high outdoor air pollution concentration, enhanced monitoring and management are required to mitigate the effect on IAQ. Future research should consider additional factors that may affect health through an understanding of the complex relationship between IAQ and subjective symptoms.

## Figures and Tables

**Figure 1 ijerph-19-02446-f001:**
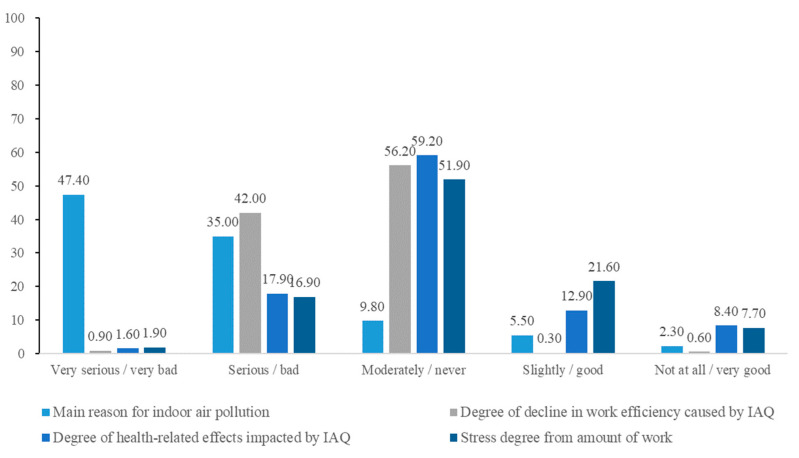
Office workers’ perception survey for diagnostic approach through questionnaire.

**Figure 2 ijerph-19-02446-f002:**
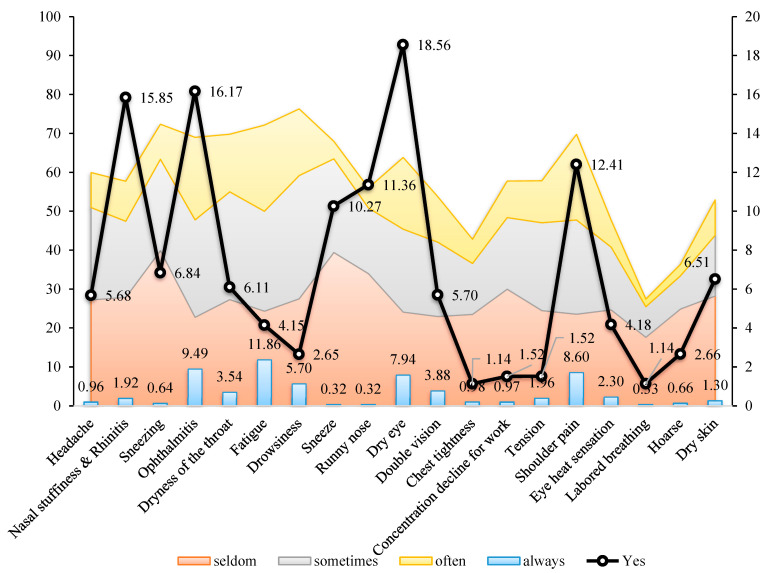
The self-diagnosis experience and pseudo-diagnosis rates for participants’ subjective symptoms.

**Table 1 ijerph-19-02446-t001:** Methods of measurements and analysis for air pollutants in offices.

Pollutants	Sampling Method and Time	Analytical Method	LOD *	Standards **
CO **	NDIR (Non-dispersive infrared absorption) Sampling time: 30 min, 2 times (30 min before and after working)	Direct-reading instrument	0.01 ppm	10 ppm
CO_2_ **				1000 ppm
HCHO **, Acetaldehyde **	2,4-DNPH Cartridge (0.5–1.0 L/m) Sampling time: 1 h ※ EPA Method TO-11 (Formaldehyde in ambient air)	HPLC (High performance liquid chromatography)	0.07 µg/m^3^ 0.05 µg/m^3^	120 µg/m^3^ (HCHO) (or 0.1 ppm)
TVOCs **	Tenax-tube (50–100 mL/m) Sampling time: 1 h ※ EPA Method TO-17 (Volatile organic compounds in ambient air)	GC-MS (Gas chromatography)	Benzene: 0.04 µg/m^3^ Toluene: 0.12 µg/m^3^ Xylene: 0.02 µg/m^3^ Ethylbenzene: 0.03 µg/m^3^	500 µg/m^3^
O_3_ ***	Coated glass filter preloaded in cassettes, nitrite impregnated (passive sampler) sampling time: 6 h	IC (Ion chromatography)	3 ppb	N.A. *****
NO_2_ ****	Badge-type diffusive sampler (passive sampler) Sampling time: 1 h	UV-VIS Spectro-photometer	6 ppb	0.1 ppm

* LOD: limit of detection. ** Reference: Air quality guidelines for offices in Korea (Ministry of Labor) and IAQ testing method and standards in Korea (Ministry of Environment). *** Reference: Korea KOSHA CODE A-1-2004 method (US OSHA, Korea KOSHA). **** Reference: Yukio Yanagisawa and Hajime Nishimura (1982). ***** N.A.: not available.

**Table 2 ijerph-19-02446-t002:** Characteristics of office workers according to subjective symptoms.

Category			Symptom Group (Mean ± SD)	Non-Symptom Group (Mean ± SD)	*p*-Value
Dry eye	Gender	Men (*n* = 199)	58.10% (*n* = 126)	83.90% (*n* = 73)	0.00
		Women (*n* = 105)	41.90% (*n* = 91)	16.10% (*n* = 14)	
	Age		37.85 ± 9.32	39.21 ± 10.04	0.27
	Hours spent indoors (h)		8.48 ± 2.71	7.51 ± 3.17	0.01
Ophthalmitis	Gender	Men (*n* = 199)	62.40% (*n* = 148)	76.10% (*n* = 51)	0.04
		Women (*n* = 105)	37.60% (*n* = 89)	23.90%(*n* = 16)	
	Age		38.00 ± 9.37	39.07 ± 10.13	0.42
	Hours spent indoors (h)		8.54 ± 2.76	7.01 ± 2.97	0.00
Nasal stuffiness & rhinits	Gender	Men (*n* = 199)	61.00% (*n* = 111)	72.10% (*n* = 88)	0.05
		Women (*n* = 105)	39.00% (*n* = 71)	27.90% (*n* = 34)	
	Age		37.69 ± 9.38	39.07 ± 9.74	0.22
	Hours spent indoors (h)		8.45 ± 2.82	7.84 ± 2.92	0.07
Headache	Gender	Men (*n* = 199)	61.30% (*n* = 114)	72.00% (*n* = 85)	0.06
		Women (*n* = 105)	38.70% (*n* = 72)	28.00% (*n* = 33)	
	Age		37.26 ± 8.85	39.79 ± 10.39	0.02
	Hours spent indoors (h)		8.72 ± 2.61	7.40 ± 3.10	0.00

**Table 3 ijerph-19-02446-t003:** Indoor air pollutant concentration according to workers’ responses about subjective symptoms.

Category	Indoor Air Pollutant	Response to Symptoms		*p*-Value
		Symptom Group (Mean ± SD)	Non Symptom Group (Mean ± SD)	
Dry eye	Benzene	4.23 ± 3.76	3.37 ± 3.36	0.05
Ophthalmitis	Benzene	4.30 ± 3.74	2.87 ± 3.12	0.02
	Ethylbenzene	13.14 ± 8.52	10.87 ± 8.91	0.05
	Acetaldehyde	13.50 ± 12.01	11.03 ± 8.14	0.05
Headache	Benzene	4.28 ± 3.76	3.52 ± 8.14	0.07

**Table 4 ijerph-19-02446-t004:** Multi-linear regression to determine factors affecting indoor air pollutant concentrations.

Category	Variable	Linear Regression	R^2^	*p*-Value
Benzene	X1: Outdoor concentration of HCHO	y = 0.088X1 + 0.040X2 + 0.081X3 + 0.020X4 –1.191	0.626	0.01
	X2: The number of photo copiers			0.05
	X3: Outdoor concentration of Benzene			0.02
	X4: Outdoor humidity			0.05
TVOCs	X1: The number of printers	y = 22.105x + 29.732	0.144	0.04
Toluene	X1: The number of photo copiers	y = 0.070x + 4.133	0.243	0.01
Ethylbenzene	X1: The number of photo copiers	y = 0.063x + 2.068	0.189	0.02
*m, p*-xylene	X1: The number of photo copiers	y = 0.052x + 1.776	0.172	0.02
*o*-xylene	X1: Outdoor concentration of HCHO	y = 0.093x + 1.020	0.187	0.02
Formaldehyde	X1: Use of air purifier	y = −1.784x + 3.576	0.341	0.00
Acetaldehyde	X1: Use of air purifier	y = −1.048x + 2.421	0.191	0.03

**Table 5 ijerph-19-02446-t005:** Health risk assessment for workers exposed to indoor air pollutants in offices.

	Air Pollutant	Cancer Risk								
		Point Estimate	Probabilistic							
		CTE *	Mean	Max	Min	Percentiles				
						25	50	75	90	95
Carcinogen	Benzene	2.9 × 10^−6^	2.91 × 10^−6^	9.10 × 10^−6^	0	1.08 × 10^−6^	1.99 × 10^−6^	3.62 × 10^−6^	6.14 × 10^−6^	8.45 × 10^−6^
	Formaldehyde	4.8 × 10^−5^	4.77 × 10^−5^	2.1 × 10^−3^	0	1.41 × 10^−5^	2.86 × 10^−5^	5.70 × 10^−5^	1.0 × 10^−4^	1.5 × 10^−4^
	Acetaldehyde	2.7 × 10^−6^	2.70 × 10^−6^	4.94 × 10^−5^	0	1.17 × 10^−6^	2.01 × 10^−6^	3.42 × 10^−6^	5.46 × 10^−6^	7.20 × 10^−6^
Non- carcinogen	Toluene	2.4 × 10^−2^	2.4 × 10^−2^	6.5 × 10^−1^	5.2 × 10^−4^	9.5 × 10^−3^	1.6 × 10^−2^	3.0 × 10^−2^	5.0 × 10^−2^	6.8 × 10^−2^
	Ethylbenzene	1.3 × 10^−2^	1.3 × 10^−2^	1.7 × 10^−1^	6.1 × 10^−4^	7.2 × 10^−3^	1.1 × 10^−2^	1.6 × 10^−2^	2.4 × 10^−2^	3.1 × 10^−2^
	*m, p*-xylene	9.2 × 10^−2^	9.2 × 10^−2^	9.2 × 10^−1^	6.7 × 10^−3^	5.2 × 10^−2^	7.2 × 10^−2^	1.1 × 10^−1^	1.6 × 10^−1^	2.0 × 10^−1^
	*o*-xylene	6.4 × 10^−2^	6.4 × 10^−2^	6.0 × 10^−1^	3.4 × 10^−3^	3.6 × 10^−2^	5.3 × 10^−2^	8.0 × 10^−2^	1.1 × 10^−1^	1.4 × 10^−1^
	NO^2^	5.0 × 10^−4^	6.1 × 10^−4^	2.1 × 10^−3^	1.0 × 10^−4^	3.8 × 10^−4^	4.8 × 10^−4^	6.0 × 10^−4^	7.3 × 10^−4^	8.3 × 10^−4^
	O^3^	4.9 × 10^−4^	4.9 × 10^−4^	1.8 × 10^−3^	1.0 × 10^−4^	3.7 × 10^−4^	4.6 × 10^−4^	5.8 × 10^−4^	7.1 × 10^−4^	8.1 × 10^−4^

* CTE: Central tendency exposure.

## Data Availability

Not applicable.

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
