# Peer review of "Risk Assessment of Indoor Air Quality and Its Association with Subjective Symptoms among Office Workers in Korea"

_ijerph, 2022, doi:10.3390/ijerph19042446_

Round 1

Reviewer 1 Report

The title addresses an important topic. However the following in addition to the intext comments on the manuscript need to be addressed 

Was calibration of the various instruments  undertaken 

What was the criteria for selecting 31 offices  and 328 workers ( line 86; 102)

Reviewer 2 Report

I read your paper with interest. However, the following should be modified for better manuscript.

1. The biggest limitation in this study is the temporal ambiguity between the time of measurement and the onset of symptoms. Also, misclassification bias might occur because workplace measurements, not individual measurements, were used as individual exposures. These limitations should be taken seriously in the discussion.

2. The number of tables is too large. The number of tables suitable for one paper is 5 or less. Tables that are not very important should be converted to supplementary tables.

3. In the text and tables, p<0.01 or p<0.05 is indicated. Instead, I recommend that indicate the p-value itself.

4. In Table 4, the Pearson correlation coefficient is a statistical method for calculating the correlation between continuous variables. Because the subjective symptom variable is a categorical variable, it is not applicable to use this statistical method.

5. There is no mention of clinical mechanism associated with indoor air pollutants (Sick Building Syndrome) in the text. I recommend that this description should be supplemented in the revision.

Round 2

Reviewer 2 Report

Most of the things mentioned have been corrected appropriately.

However, since the correlation coefficient in Table 3 is not appropriate in a statistical way, it must be corrected.

Many researchers misunderstand the Likert Scale as continuous data and use the wrong statistical method. Likert scale item is in fact a set of ordered categories, and the intervals between the scale values ​​are not equal. Any mean, correlation, or other numerical operation applied to them is invalid. Only non-parametric statistics should be used on Likert scale data (Jamieson, 2004). For correlation between a continuous and categorical variable, methods such as logistic regression and ANOVA should be used. I recommend that you delete the correlation analysis or use another appropriate method in Table 3 , and properly modify and describe the interpretation of the new analysis.

The following references are thought to be helpful. It is recommended to read and refer to.

- Grace-Martin, K. Can Likert Scale Data ever be Continuous?. Available URL: https://www.theanalysisfactor.com/can-likert-scale-data-ever-be-continuous/

- Carifio, J. & Perla, R. (2007). Ten Common Misunderstandings, Misconceptions, Persistent Myths and Urban Legends about Likert Scales and Likert Response Formats and their Antidotes. Journal of Social Sciences, 2, 106-116. Available URL: http://thescipub.com/PDF/jssp.2007.106.116.pdf

- Jamieson, S. (2004) Likert scales: how to (ab)use them? Medical Education, 38(12), pp. 1217-1218. (doi: 10.1111/j.1365-2929.2004.02012.x) Available URL: http://eprints.gla.ac.uk/59552/

- An overview of correlation measures between categorical and continuous variables. Available URL: https://medium.com/@outside2SDs/an-overview-of-correlation-measures-between-categorical-and-continuous-variables-4c7f85610365 

Author Response

->   We appreciate your thoughtful review and good comments to improve this manuscript. We have revised a point-by-point response to your comments. This manuscript has extensively been proofread by a native speaker of English by Editage (www.editage.co.kr).

Comments and Suggestions for Authors

Most of the things mentioned have been corrected appropriately.

  • We have revised a point-by-point response to your comments. Thank you again.

However, since the correlation coefficient in Table 3 is not appropriate in a statistical way, it must be corrected. Many researchers misunderstand the Likert Scale as continuous data and use the wrong statistical method. Likert scale item is in fact a set of ordered categories, and the intervals between the scale values are not equal. Any mean, correlation, or other numerical operation applied to them is invalid. Only non-parametric statistics should be used on Likert scale data (Jamieson, 2004). For correlation between a continuous and categorical variable, methods such as logistic regression and ANOVA should be used. I recommend that you delete the correlation analysis or use another appropriate method in Table 3 , and properly modify and describe the interpretation of the new analysis. The following references are thought to be helpful. It is recommended to read and refer to.

- Grace-Martin, K. Can Likert Scale Data ever be Continuous?. Available URL: https://www.theanalysisfactor.com/can-likert-scale-data-ever-be-continuous/

- Carifio, J. & Perla, R. (2007). Ten Common Misunderstandings, Misconceptions, Persistent Myths and Urban Legends about Likert Scales and Likert Response Formats and their Antidotes. Journal of Social Sciences, 2, 106-116. Available URL: http://thescipub.com/PDF/jssp.2007.106.116.pdf

- Jamieson, S. (2004) Likert scales: how to (ab)use them? Medical Education, 38(12), pp. 1217-1218. (doi: 10.1111/j.1365-2929.2004.02012.x) Available URL: http://eprints.gla.ac.uk/59552/

- An overview of correlation measures between categorical and continuous variables. Available URL: https://medium.com/@outside2SDs/an-overview-of-correlation-measures-between-categorical-and-continuous-variables-4c7f85610365.

  • Thanks you for your thoughtful review. I on behalf of co-authors agree with your opinions perfectly. In Table 3, correlation analysis between subjective symptom scores and indoor air pollutant concentrations was conducted by assigning scores to an ordinal scale (subjective symptom variable) corresponding to a qualitative variable, and converting it into a quantitative variable. As a result of the correlation analysis, the correlation showed a significantly positive. However, the correlation analysis results between the categorical variable obtained with the Likert scale and the continuous variable was not performed with an appropriate analysis method. Therefore, we deleted the correlation analyses results in Table 3. We revised the sentences in Lines 425-429.

Round 3

Reviewer 2 Report

All I mentioned have been revised properly